# Polyamine Metabolism in Scots Pine Embryogenic Cells under Potassium Deficiency

**DOI:** 10.3390/cells10051244

**Published:** 2021-05-18

**Authors:** Riina Muilu-Mäkelä, Jaana Vuosku, Hely Häggman, Tytti Sarjala

**Affiliations:** 1Natural Resources Institute Finland (Luke), 00791 Helsinki, Finland; tytti.sarjala@sci.fi; 2Ecology and Genetics Research Unit, University of Oulu, 90570 Oulu, Finland; jaana.vuosku@oulu.fi (J.V.); hely.haggman@oulu.fi (H.H.)

**Keywords:** polyamines, Scots pine (*Pinus sylvestris* L.), potassium deficiency, abiotic stress, liquid culture, chemical elements

## Abstract

Polyamines (PA) have a protective role in maintaining growth and development in Scots pine during abiotic stresses. In the present study, a controlled liquid Scots pine embryogenic cell culture was used for studying the responses of PA metabolism related to potassium deficiency. The transcription level regulation of PA metabolism led to the accumulation of putrescine (Put). Arginine decarboxylase (*ADC*) had an increased expression trend under potassium deficiency, whereas spermidine synthase (*SPDS*) expression decreased. Generally, free spermidine (Spd) and spermine (Spm)/ thermospermine (t-Spm) contents were kept relatively stable, mostly by the downregulation of polyamine oxidase *(PAO*) expression. The low potassium contents in the culture medium decreased the potassium content of the cells, which inhibited cell mass growth, but did not affect cell viability. The reduced growth was probably caused by repressed metabolic activity and cell division, whereas there were no signs of H_2_O_2_-induced oxidative stress or increased cell death. The low intracellular content of K^+^ decreased the content of Na^+^. The decrease in the pH of the culture medium indicated that H^+^ ions were pumped out of the cells. Altogether, our findings emphasize the specific role(s) of Put under potassium deficiency and strict developmental regulation of PA metabolism in Scots pine.

## 1. Introduction

Potassium (K^+^) deficiency is a common practical problem in drained peatlands in Finland. Every fourth forested stand growing on drained peatland suffers from a severe potassium shortage, and imbalances in nutrition are most common in deep-peated and nitrogen-rich sites [1]. The severe shortage may lead to dieback or even death of trees on drained peatlands [2], which enhanced further studies [3,4] on foliar nutrient diagnosis to determine the critical levels for potassium and responses to varying forest fertilization treatments. Low foliar potassium concentration in Scots pine has a slight effect on needle morphology by reducing sclerenchyma cell wall thickness and the area of the resin ducts, but increasing the area of xylem and inducing the collapse of the bundle sheath cells [5]. At the ultrastructural level, the mesophyll cells have enlarged central vacuole coinciding with an exponential increase in putrescine (Put) concentration in the needles, suggesting that the central vacuole may function as a storage site for Put [5].

Potassium is an essential nutrient and the most abundant cation in plants [6]. In plant cells, the role of potassium is related to functioning as a cofactor of enzymes and maintaining ion homeostasis by improving cell membrane stability and osmotic adjustment ability [6]. Potassium is needed in enzyme activation, protein synthesis, photosynthesis, osmoregulation, stomatal movement, energy transfer, phloem transport, cation-anion balance, and in stress resistance [7]. Due to the several functions of potassium, its intracellular content and intercellular and organ level transport are strictly regulated. 

K^+^ content of the soil is sensed by roots, and under potassium deficiency, the first response in plant cells is the hyperpolarization of membranes and decrease in extracellular pH [7]. On plasma membranes, the H^+^-ATPases produce energy that is needed to transfer the H^+^ from the cytoplasm to the apoplast leading to several parallel metabolic processes that are mainly regulated via calcium (Ca^2+^), reactive oxygen species (ROS), and phytohormones [8]. Different families of membrane-embedded ion channels regulate the K^+^, Na^+^, Ca^2+^ balance in plant cells [9]. 

Increased antioxidant capacity and changes in ion channel activity are shown to enhance the potassium deficiency tolerance of plants [10,11]. For example, transgenic tobacco (*Nicotiana tabacum* L.) plants with cationic peroxidase 1 of naturally low potassium tolerant alligator weed (*ApCPX1*) suggested that overexpression of *ApCPX1* cleared excess of ROS and depressed outward-rectifying potassium efflux channel activity [10]. In another study, the metabolite profiling of xylem sap of cotton (*Gossypium* L.) indicated that potassium deficiency reduced the antioxidant capacity of the cells, which led to damage of membranes, altered the primary and secondary metabolism, and led to the accumulation of amino acids [11].

Put accumulation under potassium deficiency was first observed in barley by Richards and Coleman [12]. Since then, potassium deficiency has been reported to cause Put accumulation in various plant species, including Scots pine (*Pinus sylvestris* L.) [3], and Norway spruce (*Picea abies* L. Karst.) [13]. In Scots pine seedlings, potassium deficiency strongly increased Put concentrations in needles, whereas spermidine (Spd) concentrations decreased in both needles and roots [4].

Polyamine (PA) accumulation is one of the most remarkable metabolic hallmarks in plants exposed to abiotic stresses, and numerous studies report improved stress tolerance achieved by either exogenous PA applications or genetic manipulation of endogenous PA levels in transgenic plants [14]. As listed by Cui et al. [8], Put seems to accumulate especially under potassium deficiency, but also under drought and osmotic stresses. However, Put content rather decreases under salt stress where the higher PAs, Spd, and spermine (Spm), may accumulate [8]. Transgenic line of Arabidopsis (*Arabidopsis thaliana* L.) with downregulated *adc2*-gene and consequent low endogenous Put content was sensitive to drought stress, which was recovered by adding exogenous Put [15]. The other study with transgenic maize (*Zea mays* L.) indicated low Put production, but high Spd/Spm content was related to improved tolerance against salt stress [16]. In potassium stress, the protective role of PAs is related to their ability to bind on plasma membranes and ion channels [17]. As positively charged molecules, PAs can bind to non-selective-cation channels (NSCC) [18] or fast (FV) or slow vacuolar (SV) channels [17] or potassium rectifying channels [19], and affect cation-anion balance in cells. The affinity of Put is lower than the affinity of Spd/Spm, which might be compensated by high Put concentration [18].

There is a connection between PA metabolism and the synthesis of abscisic acid (ABA) [20], which is one of the most common phytohormones regulating potassium, salt, drought, osmotic and other stresses in plants [8]. The arginine decarboxylase (*ADC*) gene of Arabidopsis has ABA binding domains in the promoter region, and there is a feedback loop between ABA and PA contents [21]. PAs are also involved in stress signaling in plants via the H_2_O_2_ and gamma-aminobutyric acid molecules produced as a result of PA catabolism [22,23]. Recently, the low potassium stress tolerance mechanism of transgenic tomato lines (*Solanum lycopersicum* L.) was shown to be transcriptionally regulated and mediated via a small RNA pathway, which led to changes in cytokinin and ABA signaling pathways and changes in root morphology [24]. Similarly, increased micro RNA expression and consequent Put, Spd, and Spm accumulation and antioxidant enzyme activity enhancement played a role in the root development of white pine (*Pinus strobus* L.) [25].

The different evolution of the enzyme genes in the PA biosynthesis pathway in angiosperms and gymnosperms may have resulted in increased adaptability of PA homeostasis in flowering plants compared to conifers. In Scots pine, Put is almost solely produced via the ADC pathway [26,27], unlike in most of the flowering plants, which also use ornithine decarboxylase (ODC) for Put production. There seems to be only one copy of both the *ADC* and *ODC* genes in pines [27], whereas Arabidopsis, for example, possesses two paralogues of *ADC*, which has allowed the specialization of the *AtADC1* and *AtADC2* genes [28,29]. Moreover, only one gene coding diamine oxidase (DAO), the major enzyme involved in Put catabolism, was found from the loblolly pine (*Pinus taeda* L.) genome [27], whereas *DAO* genes form large gene families in flowering plants [30]. Scots pine has a single bifunctional spermidine synthase (SPDS) enzyme for both Spd and Spm synthesis [31], whereas the duplications of *SPDS* genes has led to the presence of more than one *SPDS* genes, and further, to the evolution of separate *SPDS* and spermine synthase (*SPMS*) genes in flowering plants [32]. Furthermore, Scots pine possesses only one thermospermine synthase (TSPMS) coding *ACL5* gene [31], which in eukaryotes seems to be an ancient plant-specific gene originating from a horizontal gene transfer from Archaea or Bacteria [32]. It has been shown that Put and Spd are essential molecules for life, whereas the absence of Spm/thermospermine (t-Spm) can cause some developmental disturbances, like dwarfism, but the plant individuals are viable [33]. However, plants without Spm were stress-sensitive [34,35], indicating the role of Spm in stress protection.

Analyzing the influence of potassium deficiency on the PA metabolism in Scots pine is complicated in field environments by the occurrence of multiple simultaneous and/or successive stresses, which often occur in a correlated and interactive manner. Therefore, in the present study, we used a liquid Scots pine embryogenic cell culture as an experimental platform for studying cellular responses to potassium deficiency. As a system in which cells respond in a consistent way, minimizing variability, a liquid culture is advantageous for studying the different aspects of PA metabolism together with chemical and physiological parameters. Most importantly, the experimental design enables controlled potassium deficiency treatments without exposure of cells to other stress factors at the same time and the elimination of the possible transport of metabolites, like PAs, between plant organs or tissues. We investigated the effects of potassium deficiency on the PA metabolism in Scots pine embryogenic cells at the gene expression and metabolite levels and analyzed cell mass growth and water content, cell viability, and chemical elements. We found that transcription level regulation altered the cellular PA contents under potassium deficiency leading to the accumulation of Put like previously observed in adult Scots pine trees suffering from potassium stress [4], and young Scots pine seedling exposed to severe drought [36].

## 2. Materials and Methods

### 2.1. Establishment of Embryogenic Cell Line

One-year-old immature seed cones were collected four times from open-pollinated Scots pine *(Pinus sylvestris* L.) clone K884 in Punkaharju, Finland (61°48′ N; 29°17′ E) during summer 2004 as described in detail in Vuosku et al. [26]. Immature seeds were dissected from the developing cones, and the seed coat was removed. Embryogenic cultures were established using immature zygotic embryos with suspensor tissues surrounded by the megagametophyte. The inoculation, induction, and proliferation of the embryogenic cell masses were carried out as described in Sarjala et al. [37] on a solid basal Douglas-fir cotyledon revised (DCR) medium [38] modified by Becwar et al. [39] at room temperature in the dark. In the inoculation and induction stages, DCR medium was gelled by 4 g L ^−1^ Phytagel™ (Sigma-Aldrich Co., St Louis, MO, USA), whereas in the stage of proliferation, the amount of Phytagel™ was decreased to 2.5 g L ^−1^.

### 2.2. Liquid Cultures 

Embryogenic cells were cultured for two weeks on solid DCR medium in Petri dishes before transferring them into liquid DCR medium (pH 5.8). In the initiation of the suspension cultures, 3 g of embryogenic cell mass was mixed with 50 mL of DCR medium and transferred into 35 Erlenmeyer bottles (250 mL) that were sealed with silicone sponge closures (Sigma-Aldrich Co., St. Louis, USA). DCR media contained plant hormones cytokinin (BAP) (0.5 mg/L) and auxin (2,4-D) (0.5 mg/L) to maintain the proliferation growth. The cell suspensions were cultured at room temperature in the dark on two different shaking tables (IKA KS 260 control, IKA-WERKE GMBH and CO. KG and Heidolph Unimax 2010, Heidolph Instruments GmbH and Co. KG) with the speed of 90–110 rpm.

### 2.3. Measurement of pH and Conductivity in Culture Medium

Throughout the experiment, pH was continuously measured from two Erlenmeyer-type shake flasks with three side necks for the sensors and sampling (Glasgerätebau Ochs GmbH, Bovenden, Germany) with the SENBIT® wireless system (teleBITcomGmbH, Teltow, Germany). Prior to the experiments, the pH electrodes (autoclavable EGA 186-L Meinsberger Elektroden, Germany) were calibrated with standard pH solutions 4 and 7, after which the sensors were autoclaved (121 °C, 20 min). The SENBIT® receiver unit was set to monitor the cultures every 90 s. The data management was made with SENBIT^®^ Control software (Appendix A). In addition, pH values and the conductivity of the culture medium were determined from the filtrates using pH meter (Philips PW9422) and YSI 3200 conductivity instrument (YSI, 1700/1725 Brannum Lane, Yellow Springs OH45387 USA).

### 2.4. Potassium Treatments

The cell cultures were adapted for 14 days to the liquid medium to overcome the initial lag phase, which was caused by the sudden change in the environment. Thereafter they proceeded to the exponential growth phase. The continuous measurement of pH (see Appendix A) and the preliminary observations of cell mass growth (data not shown) were used to assess the favorable time point for the potassium deficiency/excess treatments. The cell cultures were exposed to different K concentrations by replacing the liquid DCR medium with 0% K, 50% K, 100% K, and 150% K potassium concentrations, where 100% K was the normal DCR with 4.6 mmol/L of potassium. Solutions for potassium deficiency (0% K and 50% K) and excess (150%) treatments were prepared by replacing KNO_3_ and KH_2_PO_4_ of the DCR medium with (NH_4_)_2_HPO_4_ and adjusting the medium with a combination of DCR and potassium solution to reach 0%, 50%, 100%, and 150% potassium content with an equal amount of N and P in each bottle (Appendix A). To create the different K treatments, the starting growth medium (DCR = 100% K) was replaced in two steps. First, 125 mL of new treatment medium, 0% K, 50% K, 100% K, or 150%K, was added to the bottles and incubated for 24 h on shaking tables. Thereafter, 125 mL of medium was pipetted out from the bottles, and rinsing was repeated by adding and pipetting out 150mL of treatment medium in a way that the final volume of liquid was 50mL in each bottle. The sampling was performed seven days after the treatments. The embryogenic cell mass was taken from the seven bottles per treatment (=seven biological replicates) and divided for the analyses of free and soluble conjugated PAs, cell viability, cell mass growth, and analyses of chemical elements. Four out of seven biological replicates were used for the gene expression analyses. Cell masses were collected from the culture medium by 10 s light vacuum filtering.

### 2.5. The Cell Mass Growth and Cell Viability

The fresh weight (FW) of the cell mass was weighed straight after vacuum filtering. The viability of cells was determined by the commonly used biochemical marker 2,3,5-triphenyl tetrazolium chloride (TTC) staining [40], which is based on the reduction of tetrazolium salts to red colored end products in viable cells. TTC is reduced by mitochondrial dehydrogenases, and the red color implies the mitochondrial activity of the cells [41]. The test was done according to Mikula et al. [42] with slight modifications. Two 0.2 g cell samples from each bottle were collected and washed with 1.5 mL sterilized water for 1 h. Thereafter, cells were centrifuged at 3000 g for 5 min, and the water was replaced with 1.5 mL of TTC solution (0.6% TTC in Tris buffer, pH 7.5). Cells were incubated for 24 h at 30 °C in the dark and then washed with distilled water to remove TTC. Red colored formazan was released from cells by incubating them at 85 °C in 100% ethanol for 15 min. Cells were spun down, and absorbance was measured by a spectrophotometer at 485 nm wavelength (Jenway Genova MK2 Life Science Analyser, Dunmow, Essex, UK).

### 2.6. Analysis of Chemical Elements

Samples of the harvested cell mass for the total nutrients were digested by the closed wet HNO_3_-H_2_O_2_ digestion method in a microwave (CEM MDS 2000) [43], and the elemental analysis was performed by using ICP-OES-equipment (iCAP 6500 Duo, Thermo Fisher Scientific, United Kingdom). At least 0.2 g of dried cell mass was needed for the ICP analyses, and therefore, the remaining cell mass of one to three bottles from 0% treatments were pooled together to obtain three replicate samples. From other treatments, seven replicate samples were analyzed per each. 

### 2.7. Polyamine Analysis

The PA samples, consisting of about 0.1 g of cell mass, were extracted in 5% (*w*/*v*) perchloric acid. Crude extracts for free PAs were dansylated and separated by HPLC according to Sarjala and Kaunisto [3], and soluble conjugated according to Fornalé et al. [44]. The PA concentrations were expressed as nmol g^−1^ fresh weight of cell mass. The present study does not distinguish between Spm and possible t-Spm—rather, they are both included in the same fraction and referred to as Spm/t-Spm.

### 2.8. RNA Isolation and Reverse Transcription 

The total RNA was extracted from 100 mg of cell mass using total RNA purification PureLink^TM^ Plant RNA Reagent (Invitrogen Corporation, California, USA) according to the manufacturer’s instructions. The RNA samples were treated with a rDNase set (Magherey-Nagel, Duren, Germany) at 37 °C for 10 min to eliminate contaminating genomic DNA. The amount of DNase used to produce DNA-free RNA samples was three times higher than recommended by the manufacturer. The RNA samples were purified with the NucleoSpin® RNA Clean-Up kit (Macherey-Nagel, Duren, Germany). The RNA yields were measured three times with spectrophotometric OD260 measurements, and cDNA was prepared from 1 μg of total RNA, which was reverse transcribed by SuperScript VILO^TM^ cDNA synthesis kit (Invitrogen Corporation, Carlsbad, CA, USA). 

### 2.9. Quantitative Real-Time PCR Analysis

Quantitative real-time PCR analysis (qPCR) was used for studying the expression of the arginine decarboxylase (*ADC*), spermidine synthase (*SPDS*), thermospermine synthase (*ACL5*), diamine oxidase (*DAO*), and polyamine oxidase (*PAO*) genes. In addition, the expression of the catalase (*CAT*) gene related to protection against oxidative stress, the Tat-D nuclease (*TAT-D*) gene related to programmed cell death, and the retinoblastoma related protein (*RBR*) gene related to the cell cycle progression was studied. The PCR primers for the expression studies were designed against the Scots pine gene sequences (Appendix A). The expression of the target genes was normalized by the expression of two different reference genes, ubiquinone (*UBQ*) and alfa-tubulin (*TUBA*), which had very similar expression trends. However, the UBQ gene showed more stable Cp values throughout the treatments, and therefore, the *UBQ* normalized gene expression results are presented here. The quantitative real-time PCR was performed in a 20 μL reaction mixture composed of 2 μL of cDNA, LightCycler® 480 SYBR Green I Master Mix, and 100 nM gene-specific primers. PCR amplification was initiated by incubation at 95 °C for 10 min followed by 40 cycles—30 s at 95 °C, 1 min at 58 °C, and 1 min at 72 °C. The PCR conditions were optimized for high amplification efficiency of 90% for all primer pairs used. Every PCR reaction was done in triplicate to control for the variability of PCR amplification. Gene expression results were produced with LightCycler^®^ 480 version 1.5.0.39 software. 

### 2.10. Statistical Analysis

The effects of potassium treatment on PA concentrations and gene expressions, as well as potassium content, cell mass, and cell viability, were analyzed by linear regression model [45] with lm function in R environment, version 4.0.4. [46]. Individual models were fitted for free fractions of Put, Spd, and Spm. Similar models were fitted for relative expressions of the *ADC*, *SPDS*, *ACL5*, *DAO*, *PAO*, *CAT*, *RBR*, and *TAT-D* genes. The 100% potassium treatment (100% K) was set as an intercept, and the other potassium treatments (0%, 50%, and 150%) were compared to the intercept. Before model fitting, PA concentrations and gene-expression variables were transformed to a logarithmic scale to achieve normality. After fitting in the model, the estimated regression coefficients and their 95% confidence intervals (CI) were back-transformed to the original scale. Similar models were fitted for potassium concentration, cell mass, and cell viability results, but without log-transformation. 

The correlation plot of the contents of free Put and chemical elements in Scots pine embryogenic cells under the potassium treatments was constructed in the R environment with the corrplot package [46], which is a graphical display of a correlation matrix [47].

## 3. Results

### 3.1. Effect of Potassium Deficiency on Viability and Growth of Embryogenic Cell Mass

The potassium content of the embryogenic cells linearly followed the potassium concentration of the growth medium, being on lower levels in the 0% and 50% potassium treatments compared to the 100% potassium treatment (Figure 1a, and Appendix A). The growth of the cell mass was highest under the 100% and 150% potassium treatments (approximately 12g of FW) and decreased linearly with the potassium content (Figure 1b, Appendix A). The low potassium contents, 0% and 50%, did not induce a decrease in cell viability. However, cell viability was higher under the 150% K treatment than under the other treatments (Figure 1c, Appendix A), suggesting the benefit of the potassium-enriched culture medium. Moisture content (MC%) of the cells decreased along with the potassium content, being 92.3% and 94.8% in 0% and 100% K treatments, respectively (Appendix A). The cells were shrunk under 0% K compared to the control treatment (100% K) (Figure 2). 

*CAT* expression decreased under potassium deficiency (Figure 2), which agrees with the view that in Scots pine cells, CAT protects against H_2_O_2_ accumulation and oxidative stress, especially during active metabolism [48,49]. Likewise, TAT-D, an evolutionary conserved apoptotic nuclease [50], and RBR, which inhibits cell cycle progression [51], showed decreasing gene expression under potassium deficiency (Figure 2 and Appendix A). *RBR* expression is connected with cell death processes in Scots pine zygotic embryogenesis [49]. Altogether, the expression of oxidative stress-, PCD and cell cycle-related genes was in line with the observations of cell mass growth and cell viability. The results suggested that potassium deficiency led to repressed metabolic activity and cell division, whereas there were no signs of H_2_O_2_-induced oxidative stress, RBR related cell cycle arrest, or increased cell death.

### 3.2. Content of Chemical Elements in Embryogenic Cells

In Scots pine embryogenic cells, the accumulation of Put under potassium deficiency was obvious (R^2^ = −0.88). Potassium content also correlated negatively with sulfur (S) content (R^2^ = −0.6). Instead, potassium content showed a positive correlation with the contents of sodium (Na) (R^2^ = 0.75), magnesium (Mg) (R^2^ = 0.65), and manganese (Mn) (R^2^ = 0.55) (Figure 3 and Appendix A).

### 3.3. Changes in pH and Conductivity of Culture Medium

During the cultivation, the pH value in the culture medium with embryogenic cells (Appendix A), as well as the pH value in the filtered culture medium, decreased along with the potassium contents. At the end of the cultivation, the pH value in the filtered medium was about one unit lower in the 0% K treatment than in the 100% K treatment (Table 1). The potassium content in the culture medium did not affect the conductivity, which is generally connected to uptake of inorganic ions by the cells [52].

### 3.4. Transcriptional Regulation of Put Accumulation under Potassium Deficiency

The free Put content of cells increased when the potassium content of the medium decreased, being 2.8 and 1.6 times higher under the 0% K and 50% K treatments compared to the 100% K treatment, respectively (Figure 4 and Appendix A). The results also indicated a slight accumulation of free Spd under the 0% K treatment, where the Spd content was 1.3 times higher than in the control (100% K). However, potassium treatments did not affect the content of free Spm/t-Spm (Figure 4 and Appendix A). In general, soluble conjugated PAs were about 30% of the contents of free PAs in cells and did not show any changes under the treatments (Appendix A). 

The expression trend of the Put producing enzyme gene *ADC* indicated slight upregulation under potassium deficiency; although there was no significant difference between the 100% K and 0% K treated cells (Figure 5 and Appendix A). The Put catabolizing enzyme gene *DAO* was not affected by the treatments (Figure 5 and Appendix A). Under the 0% K treatment, *SPDS* expression was about 30% lower than in the intercept (100% K), which indicated that conversion of free Put to Spd and Spm was downregulated on transcription level. Similarly, the expression of *ACL5* decreased under potassium deficiency, being ca. 60 to 70% lower under the 0% and 50% K treatments than in the 100% K-treatment, which indicated that also t-Spm production was transcriptionally downregulated rather than upregulated under potassium deficiency. Moreover, expression of *PAO* decreased under potassium deficiency, being higher under 150% and 100% than under 50% and 0% K treatments. The decreasing expression trend of *PAO* is potentially a consequence of the downregulation of Spd and Spm biosynthesis genes *SPDS* and *ACL5*, and indicates that the transcription level downregulation, rather than the PA biosynthesis upregulation, maintains PA levels under potassium deficiency (Appendix A).

## 4. Discussion

Under severe stresses, normal growth and developmental processes of plants are interrupted, and stress reactions are launched to protect cells and tissues from injuries. Due to the essential roles of PAs in both development and stress responses, it has been difficult to define the exact roles of PAs in plant stress protection, which seem to be also tissue and species-specific [14]. However, Put accumulation under potassium deficiency is considered as a general phenomenon across the plant kingdom [53,54], and Put has been suggested to be a useful biomarker to reveal potassium deficiency [8].

Put accumulation under potassium deficiency seems to have a specific adaptive role with considerable importance in conifers. The amino acid arginine is the main precursor of Put in Scots pine and generally in conifers because Put is almost solely produced via the ADC pathway [27]. It has been shown that conifers accumulate arginine in needles when exposed to mineral nutrient imbalances like potassium or phosphorus (P) deficiency, or nitrogen (N) in excess [55]. Put accumulation requires ATP, redox power (NADPH), and assimilated nitrogen, and is, therefore, metabolically “expensive” [8]. In conifers, strong accumulation of Put has been shown under potassium deficiency [3], but no effect of phosphorus concentrations on Put accumulation was found [13]. 

In the present study, we show that PA metabolism is closely related to potassium deficiency also in Scots pine embryogenic cells in which the changes in the expression of PA genes explained the accumulation of free Put. Potassium deficiency increased the expression of the *ADC* gene, which codes Put producing enzyme, whereas the expression of the Put catabolizing enzyme gene, *DAO*, was not affected. Furthermore, the expression of the aminopropyl transferase genes, *SPDS* and *ACL5,* were downregulated, indicating that Put was less converted to higher PAs under potassium deficiency. The downregulation of *PAO* expression suggested that the free Spd and Spm/ t-Spm contents were kept relatively stable by reducing their degradation. S-adenosylmethionine (SAM), once decarboxylated, provides the aminopropyl group for the synthesis of higher PAs, while SAM synthetase needs K^+^ as a cofactor [56]. Thus, transcription level downregulation of aminopropyltransferases under potassium deficiency may be related to hindered SAM synthesis, due to the lack of K^+^. Furthermore, SAM is the biosynthetic precursor for many sulfur-containing metabolites [57], which could be connected to the increased sulfur content in the cells suffering from potassium deficiency.

It has been reported that in plant cells, low potassium content is compensated by the accumulation of other cations (Na^+^, Ca^2+^, Mg^2+^) [58]. However, in Scots pine embryogenic cells, the low intracellular content of K^+^ decreased clearly the content of Na^+^ and also slightly the contents of Mn^+^ and Mg^2+^. Simultaneously, the free Put content increased, which suggested that Put accumulation could be a response to disequilibrium in cation composition and promotes Na^+^ deposition. Accumulation of Na^+^ is toxic, but even high contents of K^+^ are not harmful, and therefore, plants have developed mechanisms to prevent Na^+^ accumulation [59]. Na^+^ and K^+^ have similar physicochemical roles in cells, and especially under salt stress, accumulation of Na^+^ leads to an efflux of K^+^ without active maintenance of ion homeostasis [60]. K^+^ efflux prevention correlates with salt stress [61], and potassium deficiency tolerance in plants [10]. Accordingly, in transgenic tobacco plants with cationic peroxidase 1 *ApCPX1*-gene net K efflux rates were lower in roots compared to wild type under potassium deficiency [10]. Moreover, under K^+^ deficiency, the content of positively charged inorganic ions (Na^+^, Ca^2+^, Zn^2+^, Fe^2+^) decreased in xylem sap of cotton seedlings [11], indicating decreased uptake of cations by roots. Metabolite profiling of xylem sap indicated that a strategy of plants to maintain charge balance under potassium deficiency disturbed uptake of inorganic cations and was compensated by the accumulation of positively charged amino acids and inhibition of negatively charged amino acids [11].

In the present study, potassium deficiency led to a clear decrease in the pH of the culture medium, which indicated that H^+^ ions were pumped out of the cells. The role of Put under K^+^ deficiency seems to be related to its ability to bind into certain ion channels that regulate ion homeostasis [8]. Potassium crosses various cell membranes through K^+^ specific transport systems and membrane-embedded ion channels [6]. Put favors the H^+^ pumping across the plasma membrane that maintains the ionic balance across membranes and stabilizes depolarization of membranes under potassium deficiency [17,62]. Outward movement of H^+^ from the cytosol leads to a decrease of apoplastic pH. Scots pine embryogenic cells have large vacuoles and are highly undifferentiated without defined cellular functionalities. Little is known about the ion channels in embryogenic cells of Scots pine. Potentially, the high content of Put blocked the ion channels, i.e., FV and SV vacuolar channels and NSCC, which maintained the ratio between K^+^ and Na^+^ ions under potassium efficiency.

To extend our view of PA metabolic responses in Scots pine during different kinds of stresses, we summarized the results of the present study with our previous findings of Scots pine PA metabolism under potassium deficiency [4], osmotic [63,64], drought [36] and cold [48] stresses (Table 2). In proliferating embryogenic cells, only Spm/t-Spm content increased under polyethylene glycol (PEG)-induced osmotic stress [63]. In plants, Spm is known to improve photosynthesis and osmoregulation and enhance antioxidant defense under drought stress [65]. Furthermore, PA metabolism is also associated with somatic embryogenesis (SE) [64], and in vitro rooting in *Pinus* species [25]. When Scots pine SE was induced by ABA and the desiccation of the embryogenic cell masses by PEG, PA gene expression was downregulated on the culture medium containing both ABA and PEG, whereas PA gene expression recovered on the medium containing only ABA [64]. Thus, the transcriptional upregulation of the PA biosynthesis in the presence of ABA was suppressed under osmotic stress, which suggested that stress protection, ABA, and the transcription level regulation of PA metabolism are connected [21,66]. Furthermore, in drought-stressed Scots pine seedlings, Put content increased in needles and roots, whereas the Spd content increased only in roots [36]. Under drought stress, ABA-induced Put accumulation is involved in regulating stomatal movements by controlling inward K^+^ channels in guard cells [67], thus also forming a link between ABA, PA metabolism, and drought/osmotic stress protection [10,11]. Also, Spd has been found to increase drought tolerance in pines [68].

Altogether, the studies indicated the strict developmental regulation of PA metabolism in Scots pine cells under different stresses. The expression of both PA biosynthetic and PA catabolic genes tended to decrease rather than increase except for *ADC*. Moreover, PA contents remained relatively stable except for the Put content, which increased under potassium deficiency in embryogenic cells and the needles of adult trees, as well as in the needles and roots of drought-stressed seedlings (Table 2). Generally, the downregulation of PA catabolizing genes, especially PAO, seemed to be an important mechanism to maintain relatively stable PA contents in Scots pine cells under stress conditions. PA catabolism has been suggested to be involved in stress responses also via H_2_O_2_ production [22,23]. In cells, H_2_O_2_ is an important stress signaling molecule, but also causes oxidative stress in high concentrations [69]. Unlike in the roots of Scots pine seedlings under severe drought stress [36], *CAT* expression was not upregulated in Scots pine embryogenic cells under potassium deficiency, PEG-induced osmotic stress [63] or SE induction [64], indicating that H_2_O_2_ was not overproduced in embryogenic cells. However, the seedlings exposed to freezing temperatures indicated that one of the first responses was the upregulation of *DAO* expression [48], which could be related to the production of H_2_O_2_ and launching some signaling pathways in root cells. It was also remarkable that *ACL5*, which is connected to zygotic embryogenesis and vascular development in Scots pine [27], was most consistently downregulated under the potassium, drought, and osmotic stresses, which indicated the obvious reduction of developmental processes (Table 2). 

We conclude that in drought-stressed Scots pine seedlings, nutrient uptake was restricted, and thus, Put accumulation could also be a response to potassium deficiency or an unbalance in the K^+^/Na^+^ ratio [36]. Instead, in a liquid environment under PEG-induced osmotic stress, nutrient uptake of embryogenic cells was not limited, and no Put accumulation was detected [63] (Table 2). Drought stress has a strong influence on potassium nutrition as K^+^ mobility in the soil solution is highly dependent on the soil water potential [70,71]. Thus, drought may cause stress symptoms associated not only with low external water availability, but also with decreased nutrient uptake [72]. A close relationship between potassium nutritional status and plant drought resistance has been demonstrated [73], and potassium fertilization is beneficial to cope with water stress [70,73]. Adequate amounts of potassium can enhance the total dry mass accumulation of plants under drought stress compared to low potassium concentrations [74]. Ion homeostasis in the guard cells of needles regulates stomatal closure and reduces transpiration in seedlings [75]. An adequate K^+^ supply is also essential for enhancing drought resistance by increasing root elongation and maintaining cell membrane stability [10,73].

In conclusion, our findings emphasize the specific role(s) of Put in Scots pine embryogenic cells under potassium deficiency, and in general, strict developmental regulation of PA metabolism in Scots pine under various stresses. In Scots pine, the PA metabolism may show less flexibility under stress situations than generally in flowering plants, due to the presence of many single-copy genes and the bifunctional SPDS enzyme in the PA pathway. Furthermore, our results suggest that Put accumulation also in other stresses, such as severe drought, may be at least partly caused by potassium deficiency.

## Figures and Tables

**Figure 1 cells-10-01244-f001:**
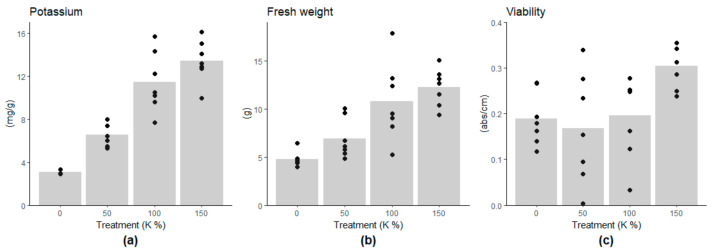
Effect of potassium treatments (0%, 50%, 100%, and 150%) on (**a**) intracellular potassium content per dry weight, (**b**) fresh weight and (**c**) viability of cells. The low potassium contents in the culture medium decreased the potassium content of the cells, which inhibited the growth linearly. Cell viability was higher in the 150% K treatment compared to the other treatments.

**Figure 2 cells-10-01244-f002:**
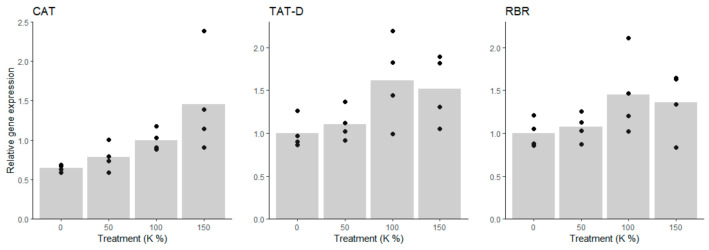
Expressions of oxidative stress-, PCD-, and cell cycle-related genes in Scots pine embryogenic cells under potassium (K) treatments. The expression of the catalase (*CAT*), Tat-D nuclease (*TAT-D*), and retinoblastoma-related protein (*RBR*) genes decreased under the 0% K treatment compared to the 100% K treatment.

**Figure 3 cells-10-01244-f003:**
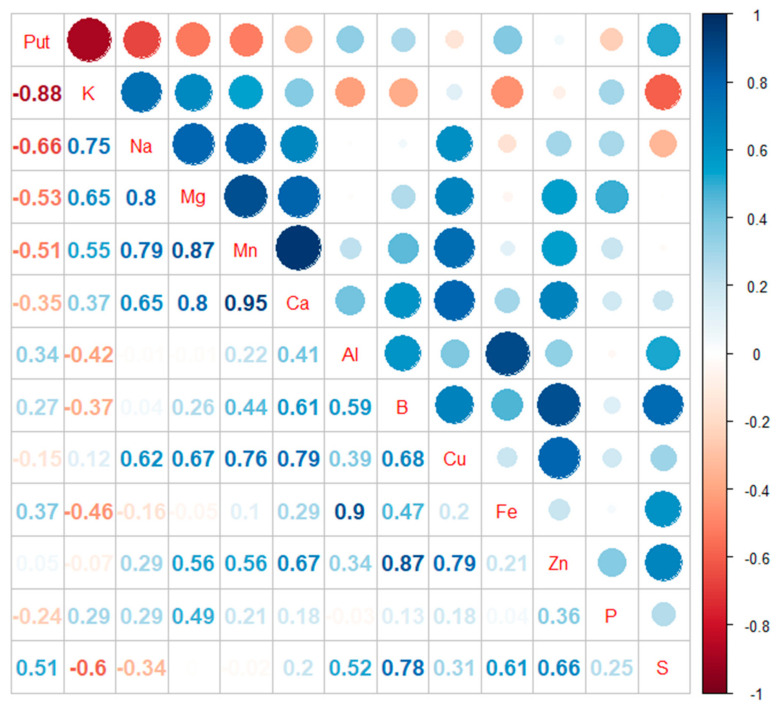
Correlation plot of the contents of free Put and chemical elements in Scots pine embryogenic cells under potassium treatments. Put correlated negatively with potassium (K, R^2^ = −0.88) sodium (Na, R^2^ = −0.66) magnesium (Mg, R^2^ = −0.53) and manganese (Mn, R^2^ = −0.51) and positively with sulfur (S, R^2^ = 0.51). Put did not show a correlation between phosphorus (P), calcium (Ca), copper (Cu), zinc (Zn), boron (B), aluminum (Al), and iron (Fe) (0.37 > R^2^ > −0.35).

**Figure 4 cells-10-01244-f004:**
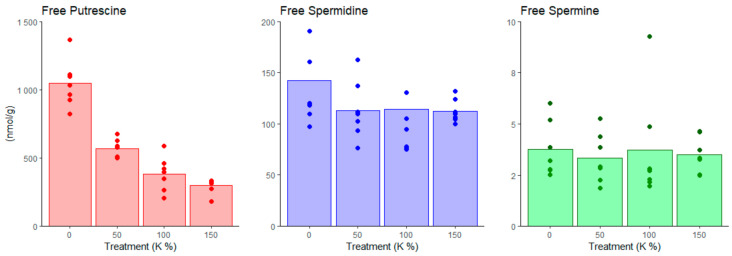
Free PA concentrations in Scots pine embryogenic cells under potassium treatments. PoTable 0. vs. 100%) increased the free Put concentration, whereas the concentrations of free Spd and Spm remained stable under the potassium treatments. PA contents were measured from the seven parallel samples (bottles) per treatment.

**Figure 5 cells-10-01244-f005:**
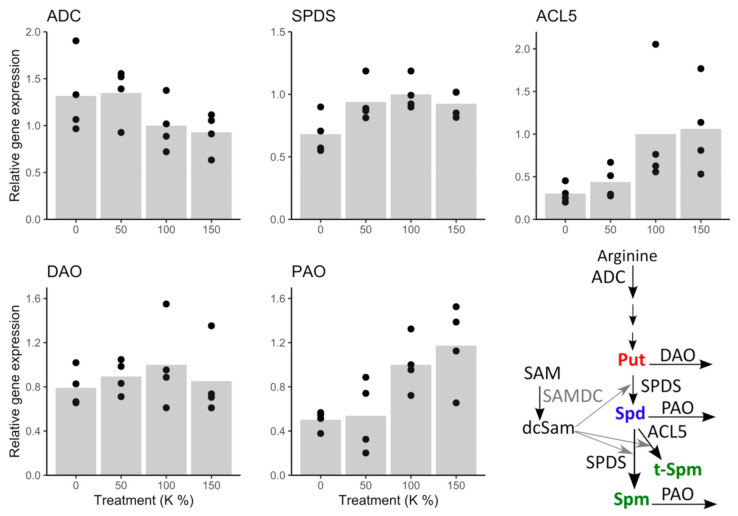
Expression of key enzyme genes in PA metabolic pathway under potassium (K) treatments. *ADC* expression showed an increasing trend, whereas the expression of *SPDS, ACL5*, and *PAO* decreased, and the expression of *DAO* was not affected by potassium deficiency.

**Table 1 cells-10-01244-t001:** pH and conductivity (Cond) of culture medium in potassium treatments.

Treatment	pH		Cond (μS/cm)	
	mean	range	mean	range
0% K	3.5	3.4–3.5	1.6	1.4–1.9
50% K	3.7	3.6–3.8	1.5	1.0–1.8
100% K	4.4	3.8–5.6	1.4	1.0–1.7
150% K	4.9	4.1–5.6	1.6	1.3–2.2

**Table 2 cells-10-01244-t002:** Summary of PA metabolism in Scots pine under potassium deficiency, drought, and cold stresses.

		PA Gene Expression					Free PA Content		
Treatment	Plant Material	ADC	SPDS	ACL5	DAO	PAO	Put	Spd	Spm/ t-Spm
Potassium deficiency in liquid cell cultures	Proembryogenic cells	**⇧**	**⇩**	**⇩**		**⇩**	**⇧**		
Potassium deficiency [4]	Needles	nd	nd	nd	nd	nd	**⇧**	**⇩**	**⇩**
Induction of osmotic stress by PEG treatment in liquid cell cultures [63]	Proembryogenic cells			**⇩**		**⇩**			**⇧**
Triggering of SE by ABA + PEG treatment [64]	Embryogenic cells	**⇩**	**⇩**	**⇩**	**⇩**	nd			
Transferring to culture medium with ABA and without PEG		**⇧**	**⇧**	**⇧**	**⇧**	nd			
Exposure of seedlings to drought stress [36]	Needles					**⇩**	**⇧**		
	Stems			**⇩**	**⇩**	**⇩**			
	Roots	**⇧**		**⇩**			**⇧**	**⇧**	
Exposure of seedlings to spring frost [48]	Needles	**⇩**	**⇩**						
	Roots				**⇧**				

## Data Availability

Data sharing is not applicable to this article.

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
