# Peer review of "Polyamine Metabolism in Scots Pine Embryogenic Cells under Potassium Deficiency"

_cells, 2021, doi:10.3390/cells10051244_

Round 1

Reviewer 1 Report

The present study reports on controlled liquid Scots pine embryogenic cell culture used for studying the responses of Polyamines(PA) metabolism related to potassium deficiency. The results highlight the specific role(s) of Putrescine(Put) in Scots pine embryo-465 genic cells under potassium deficiency and, in general, strict developmental regulation of 466 PA metabolism in Scots pine under various stresses and strict developmental regulation of PA metabolism in Scots pine.

Abstract

The abstract is written well, however there are typo errors like space (line 12-Put-Full form for first time).

Introduction

Introduction is written concisely covering latest literature available with latest citations. However, authors need to follow referencing in text as per journal guidelines. Line 28-Moilanen et al. (2010) [1], It is mixture of author date and numbering. This typo error is noticed in elsewhere in text and need to be amended before submitting revised manuscript.

Methodology

The methods given in this paper are appropriate, clear with possibility of replication. They are well designed and executed.

Results& Discussion

The data presented in an appropriate way. The Tables and Figures are appropriately presented.

Conclusion

The discussion is neatly written with all the parameters are clearly interlinked.

References

The citations used are appropriate with updated information as per journal guidelines. There is scope in the manuscript to add more 2020 references in Introduction.

Regards

Dr Suresh Naidu

Author Response

Thank you for your valuable comments. We have considered all the suggestions and revised the manuscript as suggested. The parts of the text that were revised are highlighted with yellow. We hope that the corrections can be estimated to be sufficient. Please, let us know if you find something that should be reconsidered.

Reviewer 2 Report

Comments on Manuscript ID: cells-1186457

Title: Polyamine metabolism in Scots pine embryogenic cells under potassium deficiency

General comments

The paper is well written. My only general question is the claim of use of statistic: Table 3 is the only place where the claimed “linear model with lm function and lme library in statistical program R” was applied. The rest of the data is just the row data put in the figures without standard deviation.

Detailed comments

Page 2, line 80: “The different evolution of the enzyme genes in the PA biosynthesis pathway in the two seed plant lineages may have resulted in increased adaptability of PA homeostasis in flowering plants compared to conifers.” Two seed plant lineages?

Page 4, line 170: “Thereafter, 125 ml of medium was pipetted out from the bottles and rinsing was repeated by adding and pipetting out 150ml of treatment medium in a way that the final volume of liquid was 50ml in each bottle. The sampling was performed seven days after the treatments when seven independent biological replicates (bottles) were taken from each treatment for the analyses…” Please clarify how the independent replicates were created.

Page 5, line 240: “Data was analyzed by using a linear model with lm function and lme library in statistical program R.” Fig 3 has the claimed data analysis. The rest are row data with high deviations with the claim “tendencies”.

Page 5, line 254: “The low potassium contents, 0% and 50%, did not induce a decrease in cell viability. However, cell viability was highest under the 150% K treatment indicating the benefit of the potassium-enriched culture medium.” The viability data are very wild spread in the replicates of 50 and 100% K treatments (Figure 1) so authors may be right but data is not supporting it 100%.

Page 10, line 424: My understanding that transferring to culture medium with ABA and without PEG data (Table 2) is coming from previous article as well (ref 57).

Author Response

Thank you for your valuable comments that helped us to improve our manuscript. We have considered all the comments and revised the manuscript according to the suggestions. We agree that statistics needed some clarifications and section 2.10 was revised. We have added some more recent references and revised the introduction and discussion. The changes in the text were highlighted with yellow. We hope that our revisions can be estimated to be sufficient as such. Please, let us know if there is something else that should be considered. 

Sincerely,

Riina Muilu-Mäkelä

Reviewer 3 Report

Dear Authors,

thank you for your experimental manuscript "Polyamine metabolism in Scots pine embryogenic cells under potassium deficiency"

My main concerns are as follows:

Up to date references missing (eg. 10.1186/s40064-016-2080-1; 10.1016/j.jplph.2015.08.006; 10.5511/plantbiotechnology.16.0520a; 10.1016/j.jbiosc.2020.06.007; 10.3390/cells10020261;

Add more information on the absorption pathways (potassium) of plants under stress (I recommended add: 10.1186/s12870-021-02855-4; 10.3389/fpls.2020.592591; 10.3390/ijms21072537)

The discussion section needs to be revised. Arguments require a clearer and more accurate presentation. The understanding of plant uptake mechanisms is limited because it is limited to works that have a specific view and deliberately ignore alternatives, and do not represent a balanced view of the evidence.

Ten self-citation of Vuosku J. is too much for 66 references.

The growing season needs to be clarified (l. 123)

Caption 2.10 - needs to improve - no clear significance in figures, post hoc test missing?

Table 1. better SE than the range.

Author Response

Thank you for your valuable comments that helped us to improve our manuscript. We have considered all the comments and revised the manuscript according to the suggestions. Especially, we concentrated on to revise the introduction and discussion sections according to the new information based on the suggested new references. The revised parts of the text are highlighted with yellow. Also, the statistical analyses and section 2.10 was revised. We hope that our corrections can be estimated to be sufficient as such. Let us know if there is something to be reconsidered.

Sincerely yours,

Riina Muilu-Mäkelä

Round 2

Reviewer 3 Report

Authors adressed all comments.